# Unravelling the genetic diversity of water yam (*Dioscorea alata* L.) accessions from Tanzania using simple sequence repeat (SSR) markers

**Joseph Innocent Massawe**[1,2]*, **Gladness Elibariki Temu**[2]

**1** Department of Molecular Biology and Biotechnology, University of Dar es Salaam, Dar es Salaam, Tanzania, **2** Department of Biology, University of Dodoma, Dodoma, Tanzania

* joseph.innocent92@gmail.com, joseph.massawe4@udom.ac.tz

## Abstract

Water yam *(Dioscorea alata* L.) is among the most cultivated species used as a source of food and income for small-scale farmers in Tanzania. However, little is documented about *Dioscorea* species available in Tanzania, including their genetic diversity. This study used ten polymorphic microsatellite markers to determine the genetic diversity and relationship of 63 *D. alata* accessions from six major producing regions. Results revealed a polymorphic information content (PIC) of 0.63, while the number of alleles per locus ranged from 4 to 12 with a mean of 7.60. The expected heterozygosity ranged from 0.20 to 0.76, with a mean of 0.53, which suggests moderate genetic diversity of *D. alata* accessions. Kagera region had the highest mean number of (1.5) private alleles. Analysis of molecular variance revealed that 54% of the variation was attributed to within individual, 39% was attributed to among individual while among population contributed 7% of the total variation. The highest Nei's genetic distance (0.43) was for accessions sampled from Kilimanjaro and Mtwara regions. Principal coordinate analysis and cluster analysis using Unweighted Paired Group Method using Arithmetic (UPGMA) grouped *D. alata* accessions into two major clusters regardless of geographical origin and local names. The Bayesian structure analysis confirmed the two clusters obtained in UPGMA and revealed an admixture of *D. alata* accessions in all six regions suggesting farmers' extensive exchange of planting materials. These results are helpful in the selection of *D. alata* accessions for breeding programs in Tanzania.

## Introduction

Yam (*Dioscorea* spp.) is an important food security tuber crop supporting more than 300 million people worldwide [1]. The crop is mainly grown in the tropical and subtropical regions of the world, especially in the yam belt of West African countries [2, 3]. In countries where yam is highly cultivated, it is a staple food due to its high carbohydrate (about 77.5% of dry matter) and other mineral contents [4]. The genus *Dioscorea* contains more than 650 species whereby 10 being cultivated while many others are wild species [5].

Water yam (*Dioscorea alata*), also known as greater or winged yam, is a dioecious and polyploid *Dioscorea* species with a ploidy level (2n = 40 to 80) [6]. The primary chromosome

**Data Availability Statement:** All SSR data are available from the Dryad database. URL: https://datadryad.org/stash/dataset/doi:10.5061/dryad.

pg4f4qrv4 DOI: https://doi.org/10.5061/dryad.
pg4f4qrv4.

**Funding:** JIM received the research fund under the MoEST 2018 sponsorship. This study was funded by Ministry of Education, Science and Technology (Tanzania). The funder can be accessed via https://www.moe.go.tz/sw. The funders had no role in study design, data collection and analysis, decision to publish, or preparation of the manuscript.

**Competing interests:** The authors have declared that no competing interests exist.

number of *D. alata* is 20 [6, 7]. *D. alata* ranks second in production after Guinea yam Cormier et al. [8], although it is the most widely distributed yam globally [9]. *D alata* has high importance in food security than other cultivated yam species due to its high yield potential, tuber storability and ease of propagation due to the production of bulbils [10].

Assessing yam genetic diversity is essential for germplasm management, conservation and planning for the development of new varieties [11]. The level of genetic diversity in *D. alata* can be assessed using various molecular markers. Using Random Amplified Polymorphic DNA (RAPD) and Inter Simple Sequence Repeat (ISSR) markers, Rao et al. [12] reported high variability among *D. alata* genotypes. The genetic diversity of *D. alata* accessions from West and Central Africa and Puerto Rico was assessed using Amplified Fragment Length Polymorphism (AFLP) markers and the accessions clustered into three groups, irrespective of the geographical pattern as reported by Egesi et al. [13]. Using Single Nucleotide Polymorphism (SNP) markers, Agre et al. [14] established the genetic diversity of 100 *D. alata* accessions and reported three genetic groups. Simple Sequence Repeat (SSR) markers are also among the most commonly used markers in the diversity analysis of crops, including yams. Despite being a relatively old method, SSR is less expensive than advanced techniques such as Diversity Array Technology (DarT) and Next-generation Sequencing (NGS). SSR markers are also highly preferred due to their reproducibility, transferability among related species and co-dominance nature of inheritance [15]. Using 24 SSR markers, Arnau et al. [16] established the genetic diversity and population structure of 384 *D. alata* accessions from the South Pacific, Asia, Africa and the Caribbean and revealed wide genetic diversity in six populations. Chen et al. [17] estimated the genetic diversity of 26 *D. alata* landraces from China using 9 SSR markers and reported four clusters among yam landraces.

In Tanzania, small-scale farmers mainly produce yam for food, income generation and medicinal purposes. The major yam-producing areas in Tanzania are considered to be Mtwara, Lindi, Morogoro, Arusha, Kilimanjaro, Kagera, Coastal, Zanzibar Islands Mbeya and Ruvuma regions. However, despite the importance of yam to especially farmers, there is limited research on this crop. The genetic diversity of yam accessions grown in the country has never been estimated and documented. Since water yam is the most widely cultivated species in the country, the genetic diversity of 63 *D. alata* accessions from major yam-growing regions was estimated using ten polymorphic SSR markers. This information is crucial to understanding the genetic structure of *D. alata* accessions, which will enhance management, conservation and planning for use in breeding programs in Tanzania.

## Materials and methods

### Ethical statement

This study involved collection of plant materials (yam tubers) from five regions in Tanzania. Research work, including field work and the collection of yam accessions, was permitted by the University of Dar es Salaam, Tanzania, with the informed written permit number AB3/12 (B). The respective administrative departments in each region approved collection of yam accessions.

### Plant materials

Research work, including field work and the collection of yam accessions, was permitted by the University of Dar es Salaam, Tanzania, with the informed written permit number AB3/12 (B). The 63 *D. alata* tuber accessions were collected from farmer's fields in five major growing regions in Tanzania during harvesting season between August and October 2019 (Fig 1 and Table 1). Tubers were maintained for 2 months to break the dormancy and planted to collect

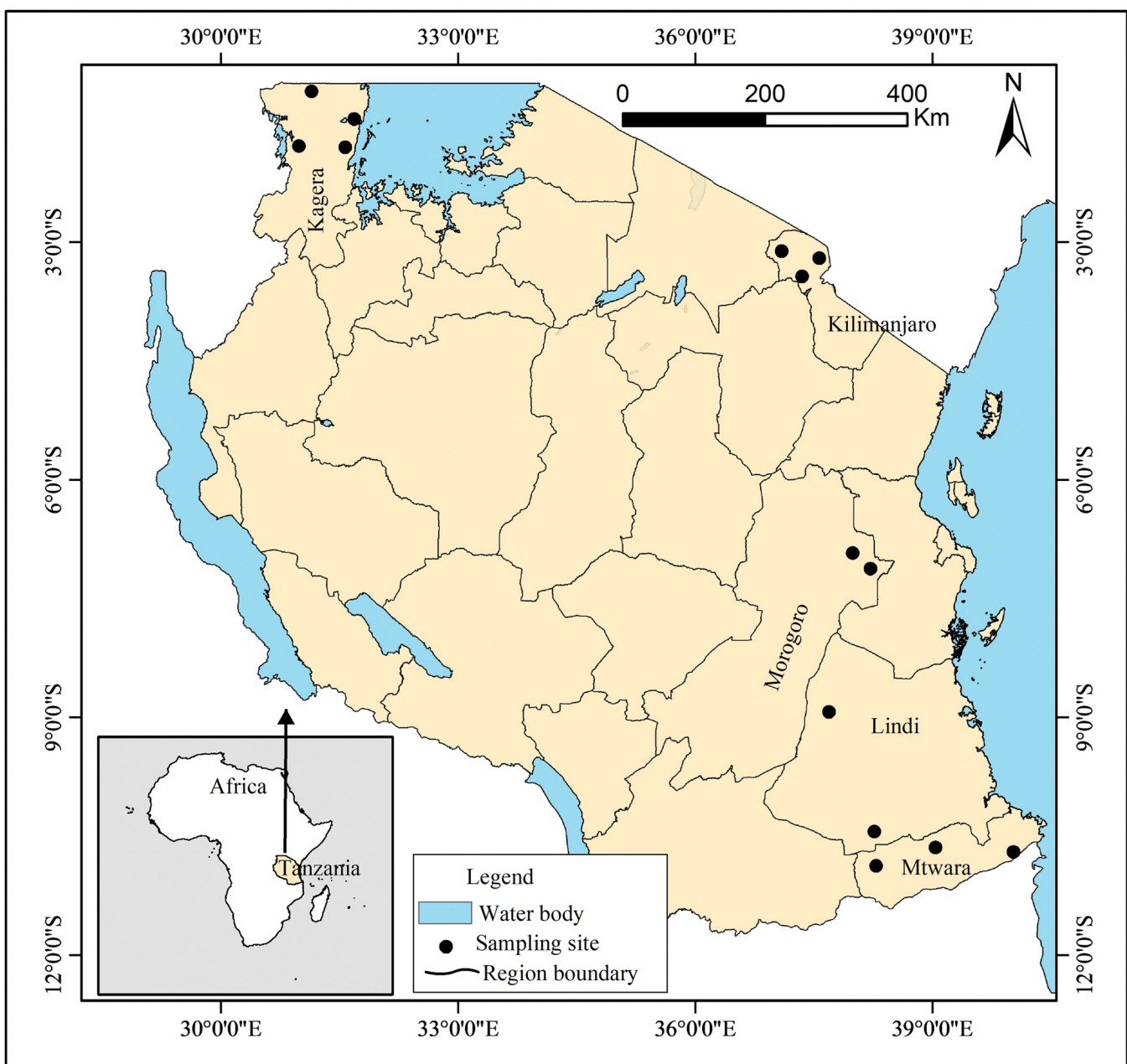

**Fig 1. Map of Tanzania showing regions and study sites where yam samples were collected.**

leaves for DNA extraction. Large tubers were sliced into minisett of about 60 grams and planted on ridges at Tanzania Agricultural Research Institute (TARI-Kibaha) experimental plots. Standard cultural agronomic practices, including hand weeding and stacking were employed. Plants were manually irrigated once per day in three times a week. Three young fresh leaves were collected (from the replicates of each accession) two months post planting in a zip bag and transported to the International Institute of Tropical Agriculture—Tanzania for DNA extraction.

**Table 1. Description of 63 *D. alata* accessions used in this study.**

| S/n | Accession number | Local name | Distinct morphological character | Region |
|---|---|---|---|---|
| 1 | YM1 | Kiraira | Winged stem | Kagera |
| 2 | YM2 | Kanyanyinyi | Winged stem | Kagera |
| 3 | YM3 | Kiraira | Winged stem | Kagera |
| 4 | YM4 | Kiraira | Winged stem | Kagera |
| 5 | YM5 | Kanyanyinyi | Winged stem | Kagera |
| 6 | YM6 | Kiraira | Winged stem | Kagera |
| 7 | YM7 | Kiraira | Winged stem | Kagera |
| 8 | YM8 | Kiraira | Winged stem | Kagera |
| 9 | YM9 | Kanyanyinyi | Winged stem | Kagera |
| 10 | YM10 | Kanyanyinyi | Winged stem | Kagera |
| 11 | YM11 | Kanyanyinyi | Winged stem | Kagera |
| 12 | YM12 | Kanyanyinyi | Winged stem | Kagera |
| 13 | YM13 | Kanyanyinyi | Winged stem | Kagera |
| 14 | YM14 | Ifure | Winged stem | Kilimanjaro |
| 15 | YM15 | Ifure | Winged stem | Kilimanjaro |
| 16 | YM16 | Ifure | Winged stem | Kilimanjaro |
| 17 | YM17 | Ifure | Winged stem | Kilimanjaro |
| 18 | YM18 | Ifure | Winged stem | Kilimanjaro |
| 19 | YM19 | Ifure | Winged stem | Kilimanjaro |
| 20 | YM20 | Kijabo | Winged stem | Lindi |
| 21 | YM21 | Mapeta | Winged stem | Lindi |
| 22 | YM22 | Nkwande | Winged stem | Lindi |
| 23 | YM23 | Mapeta | Winged stem | Lindi |
| 24 | YM24 | Katuri | Winged stem | Lindi |
| 25 | YM25 | Vijabo | Winged stem | Lindi |
| 26 | YM26 | Vijabo | Winged stem | Lindi |
| 27 | YM27 | Vijabo | Winged stem | Lindi |
| 28 | YM28 | Mapeta | Winged stem | Lindi |
| 29 | YM29 | Mikirachi | Winged stem | Lindi |
| 30 | YM30 | Mapeta | Winged stem | Lindi |
| 31 | YM31 | Vigonzo | Winged stem | Morogoro |
| 32 | YM32 | Msagala | Winged stem | Morogoro |
| 33 | YM33 | Vigonzo | Winged stem | Morogoro |
| 34 | YM34 | Msagala | Winged stem | Morogoro |
| 35 | YM35 | Mgendagenda | Winged stem | Morogoro |
| 36 | YM36 | Mgendagenda | Winged stem | Morogoro |
| 37 | YM37 | Mgendagenda | Winged stem | Morogoro |
| 38 | YM38 | Mgendagenda | Winged stem | Morogoro |
| 39 | YM39 | Mgendagenda | Winged stem | Morogoro |
| 40 | YM40 | Mnangilangi | Winged stem | Mtwara |
| 41 | YM41 | Mnangilangi | Winged stem | Mtwara |
| 42 | YM42 | Vitungula | Winged stem | Mtwara |
| 43 | YM43 | Mnangilangi | Winged stem | Mtwara |
| 44 | YM44 | Mnangilangi | Winged stem | Mtwara |
| 45 | YM45 | Hamandeke | Winged stem | Mtwara |
| 46 | YM46 | Hamandeke | Winged stem | Mtwara |
| 47 | YM47 | Hamandeke | Winged stem | Mtwara |

(*Continued*)

**Table 1.** (Continued)

| S/n | Accession number | Local name | Distinct morphological character | Region |
|---|---|---|---|---|
| 48 | YM48 | Mnangilangi | Winged stem | Mtwara |
| 49 | YM49 | Hangadi shamba | Winged stem | Mtwara |
| 50 | YM50 | Mnangilangi | Winged stem | Mtwara |
| 51 | YM51 | Hangadi shamba | Winged stem | Mtwara |
| 52 | YM52 | Nyuvele | Winged stem | Mtwara |
| 53 | YM53 | Nyuvele | Winged stem | Mtwara |
| 54 | YM54 | Nyuvele | Winged stem | Mtwara |
| 55 | YM55 | Ihumihumi | Winged stem | Mtwara |
| 56 | YM56 | Nyuvele | Winged stem | Mtwara |
| 57 | YM57 | Ihumihumi | Winged stem | Mtwara |
| 58 | YM58 | Mkonga | Winged stem | Mtwara |
| 59 | YM59 | Mikirachi | Winged stem | Mtwara |
| 60 | YM60 | Mkonga | Winged stem | Mtwara |
| 61 | YM61 | Vitungula | Winged stem | Mtwara |
| 62 | YM62 | Mapeta | Winged stem | Mtwara |
| 63 | YM63 | Mapeta | Winged stem | Mtwara |

## DNA extraction and quantification

Genomic DNA was extracted from 300 mg of fresh leaves following the cetyltrimethylammonium bromide (CTAB) protocol described by Allen et al. [18]. The quality of DNA was visualized by electrophoresis on a 1% agarose gel. The concentration of DNA was estimated using a spectrophotometer at 260 nm wavelength, and DNA was diluted to obtain a working concentration of 25 ng/μL.

## Polymerase chain reaction (PCR) and fragment analysis

A total of 15 SSR markers were screened using all 63 *D. alata* samples used in this study (Table 2). However, only 10 SSR markers were polymorphic and therefore were included in the analysis. PCR was carried out in a total volume of 15 μL containing 25 ng/μL of genomic DNA, 0.2 μM of forward and reverse primers and 7.5 μL of OneTaq Quick-Load 2X Master Mix with Standard Buffer (New England Biolabs, Massachusetts, USA). The PCR program was as follows; denaturation at 94˚C for 4 min followed by 35 cycles of 94˚C for 30 sec, annealing temperature (various as shown in Table 2) and 72˚C for 1 min. The final extension was held at 72˚C for 7 min. To prove amplification, 3 μL of the PCR products were run on 1.5% agarose gel (aMReSCO, Solon, Ohio, USA). The PCR products were then sent to Bioscience eastern and central Africa (BecA) Hub in Nairobi, Kenya and were separated on the ABI-3730 capillary electrophoresis (Applied Biosystems). Data was captured and the resulting fragments were scored using GeneMapper V6 software.

## Data analysis

**Genetic diversity analysis.** Genetic diversity parameters including the number of alleles per locus, number of polymorphic alleles, number of effective alleles, Shannon's Information Index, observed heterozygosity, gene diversity, inbreeding coefficient and principal coordinate analysis were determined using GenAlEx software version 6.503 described by Peakall et al. [22]. Polymorphic Information Content (PIC) was estimated using PowerMarker software V3.25 by Liu [23].

**Table 2. Description of SSR markers used to genotype 63 *D. alata* accessions collected from Tanzania.** Markers marked with an asterisk (*) were not used in the analysis.

| SN | Marker | Forward primer (5'–3') | Reverse primer (5'–3') | Motif | Ta |
|---|---|---|---|---|---|
| 1 | Da1A01 | TATAATCGGCCAGAGG | TGTTGGAAGCATAGAGAA | $(GT)_8$ | 51 |
| 2 | Dab2C05 | CCCATGCTTGTAGTTGT | TGCTCACCTCTTTACTTG | $(GA)_{19}$ | 51 |
| 3 | Dab2D08 | ACAAGAGAACCGACATAGT | GATTTGCTTTGAGTCCTT | $(AG)_{16}$ | 51 |
| 4 | Dab2E07 | TTGAACCTTGACTTTGGT | GAGTTCCTGTCCTTGGT | $(CT)_{23}$ | 51 |
| 5 | Dpr3D06 | ATAGGAAGGCAATCAGG | ACCCATCGTCTTACCC | $(GA)_{15}$ | 51 |
| 6 | Dpr3F12 | TCCCCATAGAAACAAAGT | TCAAGCAAGAGAAGGTG | $(GA)_{16}$ | 51 |
| 7 | Da1F08 | AATGCTTCGTAATCCAAC | CTATAAGGAATTGGTGCC | $(TG)_{13}$ | 51 |
| 8 | H2 | AAACCAAACAGGCAAAGCAT | TGCCCTGCTTGTAAGATTGA | $(CA)_9$ | 56 |
| 9 | F1 | ATGGCTCAAGAGCACACG | GGGCCTCATAAACATGCAAT | $(TA)_5$ | 60 |
| 10 | YM80 | CCGCCCAATCACATCACATC | TCCCAAGAAGTCTGAGCCG | $(CTT)_{13}$ | 60 |
| 11 | A4* | TTCGTTCTCGATAGCGGACT | CCAGTTCCCAGCCTCTTGT | $(CT)_2(GAA)_3GA(GAA)_3$ | 60 |
| 12 | YM13* | CCAATCACATCACGTCTAGTCT | GACAATAGAAACTTCGAGACCC | $(CTT)_8$ | 60 |
| 13 | H12* | TTGTAATTGGGTGGTTGTATTTGC | CGGCCAAAACATTTTCTGAT | $(AT)_6$ | 56 |
| 14 | Dpr3F04* | AGACTCTTGCTCATGT | GCCTTGTTACTTTATTC | $(AG)_{15}$ | 51 |
| 15 | Da1D08* | GATGCTATGAACACAACTAA | TTTGACAGTGAGAATGGA | $(CA)_8$ | 51 |

Source: [19–21]

## Analysis of molecular variance and cluster analysis

To assess the diversity level and genetic relationship among the *D. alata* population, Analysis of Molecular Variance (AMOVA) was estimated using GenAlEx software version 6.503 described by Peakall et al. [22]. The Unweighted Paired Group Method using arithmetic Average (UPGMA) was used to construct the dendrogram in PowerMarker software V3.25 by Liu [23]. The dendrogram was then generated in Molecular Evolutionary Genetics Analysis (MEGA-X) V10.1.8 described by Kumar et al. [24].

## Structure analysis

Bayesian analysis using Structure software V2.3.4 described by Pritchard et al. [25] was used to estimate the population's genetic structure. The admixture model for K values from 1 to 10 with a burn-in period of 100 000 steps and 100 000 interactions MCMC (Markov Chain Monte Carlo) was used in the analysis. The most likely optimal number of K clusters was estimated using the ad hoc parameter ($\Delta K$) method described by Evanno [26] in a Structure Harvester [27].

## Results

### Characteristics of the SSR markers

The Polymorphic Information Content (PIC) values of the ten SSR markers used ranged from 0.33 for the marker F1 to 0.85 for the marker Da1A01, with a mean of 0.63 (Table 3). The high mean PIC obtained implies that the SSR markers used in our study were very informative with high discriminating power. Hence, these markers can be used in genetic diversity analysis.

### Genetic diversity of *D. alata*

Genetic diversity parameters are presented in Table 3. The total number of alleles detected per locus was 76, ranging from 4 to 12, with a mean of 7.60. The lowest numbers of alleles per

**Table 3. Summary of genetic parameters of ten SSR markers used in assessing the genetic diversity of 63 *D. alata* accessions.**

| Locus | No. of Alleles | Ne | Ho | He | $F_{IS}$ | PIC |
|---|---|---|---|---|---|---|
| Da1A01 | 9 | 4.11 | 0.78 | 0.74 | -0.05 | 0.85 |
| Da1F08 | 6 | 3.58 | 0.70 | 0.70 | 0.01 | 0.70 |
| Dab2C05 | 7 | 1.40 | 0.19 | 0.23 | 0.19 | 0.44 |
| Dab2D08 | 7 | 2.96 | 0.69 | 0.63 | -0.09 | 0.71 |
| Dab2E07 | 8 | 4.20 | 0.96 | 0.76 | -0.27 | 0.85 |
| Dpr3D06 | 6 | 1.86 | 0.03 | 0.46 | 0.93 | 0.55 |
| Dpr3F12 | 12 | 2.74 | 0.41 | 0.62 | 0.34 | 0.69 |
| F1 | 4 | 1.31 | 0.07 | 0.20 | 0.68 | 0.33 |
| H2 | 5 | 1.67 | 0.05 | 0.37 | 0.86 | 0.48 |
| YM80 | 12 | 2.82 | 0.33 | 0.53 | 0.38 | 0.67 |
| Mean | 7.6 | 2.67 | 0.42 | 0.53 | 0.29 | 0.63 |

Ne = Number of effective alleles, I = Shannon's diversity index, Ho = Observed heterozygosity, He = Expected heterozygosity, $F_{IS}$ = Inbreeding coefficient and PIC = Polymorphic Information content.

locus were detected in marker F1 while the highest was detected in markers Dpr3F12 and YM80. The effective number of alleles per locus ranged from 1.31 to 4.20, with a mean of 2.67. The F1 and Dab2E07 markers had the lowest and highest number of effective allele locus per locus, respectively. The expected heterozygosity ranged from 0.20 to 0.76 with a mean of 0.53, whereby the lowest and highest expected heterozygosity was detected in markers F1 and Dab2E07, respectively. The inbreeding coefficient ranged from -0.27 to 0.93, with a mean of 0.29. Markers Dab2E07 and Dpr3D06 had the lowest and highest inbreeding coefficient, respectively. Three markers (30%) showed a negative inbreeding coefficient, indicating an excess of heterozygosity.

## Genetic diversity of *D. alata* based on population

The genetic diversity among and within 63 *D. alata* accessions based on the population is presented in Table 4. The number of effective allele was lowest (2.33) in Morogoro region and highest (3.32) in Kagera region, with a mean of 2.67. Shannon Information Index was lowest (0.84) in the Morogoro region and highest (1.22) in Kagera region, with a mean of 1.00. The mean number of private alleles ranged from 0.20 (Lindi region) to 1.5 (Kagera region). Observed heterozygosity (Ho) ranged from 0.32 to 0.55 in Mtwara and Kagera regions, respectively, with a mean of 0.42. The expected heterozygosity (He) ranged from 0.45 to 0.60 in Morogoro and Kagera regions, respectively, with a mean of 0.53. Our results indicate that *D. alata*

**Table 4. Genetic diversity of 63 *D. alata* accessions based on populations as generated by 10 SSR markers.**

| Population | Ne | I | Private allele | Ho | He | % Poly loci |
|---|---|---|---|---|---|---|
| Kagera | 3.32 | 1.22 | 1.50 | 0.55 | 0.60 | 100.00 |
| Kilimanjaro | 2.41 | 0.90 | 0.40 | 0.41 | 0.50 | 80.00 |
| Lindi | 2.47 | 0.95 | 0.20 | 0.42 | 0.50 | 90.00 |
| Morogoro | 2.33 | 0.84 | 0.30 | 0.41 | 0.45 | 90.00 |
| Mtwara | 2.79 | 1.14 | 0.70 | 0.32 | 0.57 | 100.00 |
| Mean | 2.67 | 1.00 | | 0.42 | 0.53 | 92.00 |

Ne = Number of effective alleles, I = Shannon diversity index, Ho = Observed heterozygosity, He = Expected heterozygosity and % Poly Loci = Percentage of polymorphic loci.

**Table 5. Analysis of Molecular Variance (AMOVA) of the SSR markers among and within 63 *D. alata* accessions.**

| Source | DF | SS | MS | Estimated Variance | Percentage variation | *p* |
|---|---|---|---|---|---|---|
| Among pops | 4 | 38.98 | 9.74 | 0.22 | 7% | 0.001 |
| Among individual | 58 | 259.50 | 4.47 | 1.32 | 39% | |
| Within individual | 63 | 116.00 | 1.84 | 1.84 | 54 | |
| Total | 125 | 414.48 | | 3.38 | 100% | |

F—Statistics
$F_{ST} = 0.07$

DF = Degree of freedom, SS = Sum of squares and MS = Mean sum of square, $F_{ST}$ = Genetic differentiation index

accessions collected from Kagera region had the highest genetic diversity followed by Mtwara region. The mean polymorphic loci were 92.00%.

## Analysis of molecular variance (AMOVA)

Analysis of molecular variance showed a highly significant difference (p < 0.001) in *D. alata* accessions within regions. The analysis showed that 54% of the variation was due to within individual, while 39% was due to among individual while only 7% of the total variation was among the population (Table 5). The genetic differentiation index ($F_{ST} = 0.06$) indicates low genetic differentiation among regions.

## Principal coordinate analysis (PCoA)

The first three principal coordinate axes accounted for 41.19% of the total variation. The PCoA of the correlation between the genetic relationship of *D. alata* accessions and geographical distribution showed that *D. alata* accessions were divided into two groups (Fig 2). Group A contained *D. alata* accessions collected from all six regions, while group B had accessions from Kagera, Morogoro and Lindi regions. The scatter plot grouped *D. alata* accessions regardless of their geographical origin and local names.

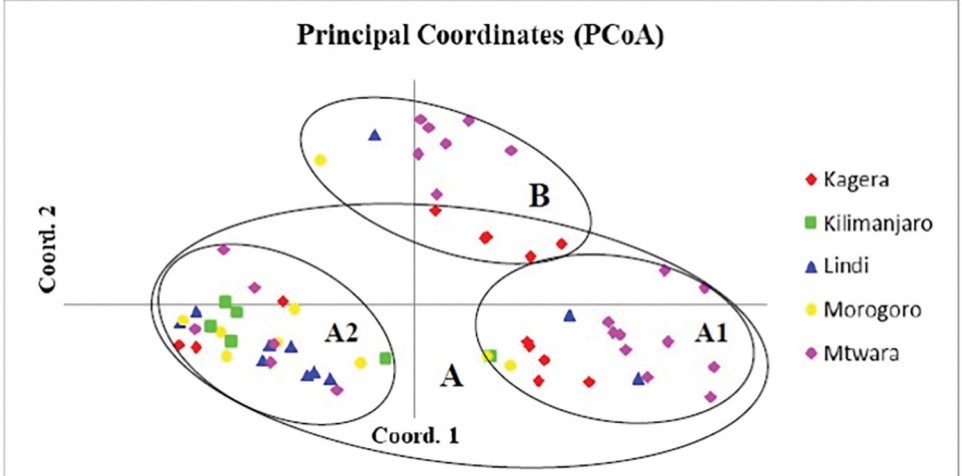

**Fig 2. Principal coordinate analysis (PCoA) showing the relationship among 63 *D. alata* accessions using 10 SSR markers.** Group A contained accessions collected from all five regions while group B had accessions from Kagera, Lindi and Morogoro. Coord. 1 and Coord. 2 represent the first and second coordinates respectively.

**Table 6. Nei's Unbiased Genetic identity (above diagonal) and genetic distance (lower diagonal) of 63 *D. alata* accessions using ten SRR markers.**

| Location | Kagera | Kilimanjaro | Lindi | Morogoro | Mtwara |
|---|---|---|---|---|---|
| Kagera | | 0.71 | 0.73 | 0.72 | 0.72 |
| Kilimanjaro | 0.34 | | 0.74 | 0.77 | 0.65 |
| Lindi | 0.31 | 0.30 | | 0.99 | 0.90 |
| Morogoro | 0.32 | 0.26 | 0.01 | | 0.83 |
| Mtwara | 0.32 | 0.43 | 0.10 | 0.19 | |

## Genetic identity and genetic distance

The Nei's unbiased genetic identity and genetic distances are presented in Table 6. The genetic distance among accessions and regions varied from 0.01 to 0.43. The highest genetic distance was for accessions collected from Kilimanjaro and Mtwara (0.43), followed by Kagera and Kilimanjaro (0.34), while the lowest genetic distance was for accessions from Lindi and Morogoro (0.01). Genetic identity between accessions and regions varied from 0.65 to 0.99. The highest genetic identity (0.99) was for accession from Morogoro and Lindi regions, while the lowest (0.52) was for accessions collected from Mtwara and Kilimanjaro.

## Cluster analysis

UPGMA grouped the 63 *D. alata* accessions into two major clusters, A and B (Fig 3). Generally, the accessions clustered regardless of their geographical origin. Clusters A and B had 49 and 14 accessions, respectively. Cluster A was further subdivided into two sub-clusters (A1 and A2). Sub-cluster A1 had 28 accessions collected from all regions surveyed, while sub-cluster A2 had 21 accessions collected from Kagera, Morogoro, Lindi and Mtwara regions. Cluster B had two sub-clusters (B1 and B2). Sub-cluster B1 contained 5 accessions from the Kagera region, while sub-cluster B2 had 9 accessions collected from Morogoro, Lindi and Mtwara regions. In group B1, accessions collected from Kagera region clustered together, suggesting a distant relationship from the rest. Results from cluster analysis confirmed the results obtained in PCoA analysis. Two clusters with sub-clusters were observed in both analyses.

## Structure analysis

The Bayesian analysis performed in STRUCTURE software for the SSR data confirmed the two clusters obtained in the UPGMA. Evanno's method showed a ΔK peak value of K = 2, suggesting the presence of two distinct clusters, which confirmed that the *D. alata* accessions collected from the six regions are genetically structured in two groups (Fig 4). The STRUCTURE results are consistent with the UPGMA cluster in that all regions are admixed, and only Kilimanjaro region had a nearly homogeneous population.

## Discussion

Estimating the genetic diversity of yam is very important in understanding the extent of genetic variation available for proper germplasm management and planning for breeding programs [11, 28].

SSR markers used in our study were highly polymorphic, suggesting that these markers were able to reveal variations within the accessions. PIC is an essential feature in evaluating the informativeness of a molecular marker [29]. Generally, the PIC values that range between 0.5 and 1.0 implies high informative markers, whereas values less than 0.5 indicate narrow informative marker [30]. Siqueira et al. [31], reported high PIC (0.92) using 11 SSR markers in 89

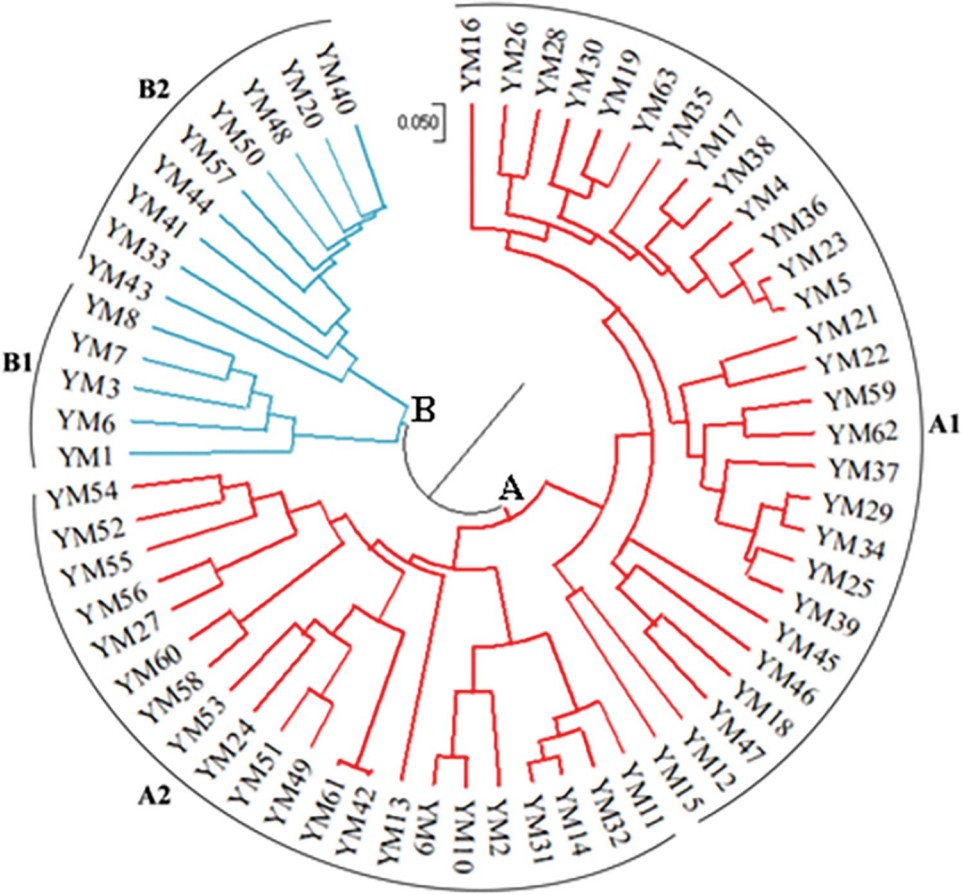

**Fig 3. UPGMA cluster analysis of 63 *D. alata* accessions based on 10 SSR markers, using MEGA-X software.** A1 and A2 are sub-clusters of cluster A while B1 and B2 represent sub-clusters of cluster B.

*D. alata* accessions collected from Brazil. Otoo et al. [32] reported high PIC (0.91) in 14 SSR markers used to assess 49 *D. alata* accessions collected from Ghana. Girma et al. [33] reported a lower mean PIC (0.43) using 8 SRR markers in 127 *D. alata* accessions from the International Institute of Tropical Agriculture (IITA) geneBank. Compared to this study, the different PIC levels observed in other studies could be attributed to different SSR markers employed, which target different loci and also the composition of the yam genotypes.

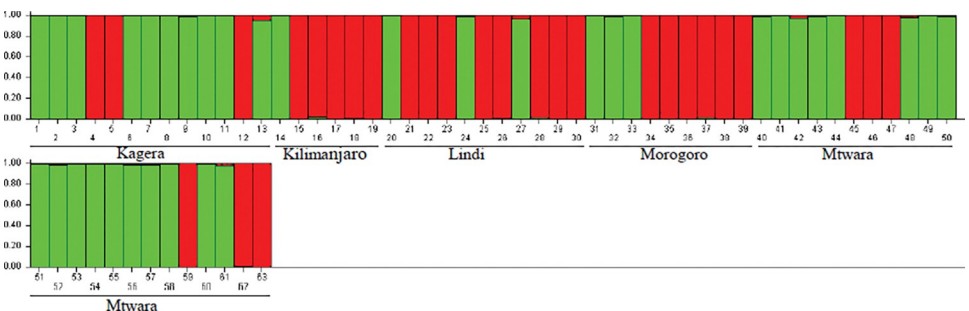

**Fig 4. Structure of the genetic diversity of 63 *D. alata* accessions based on Bayesian analysis of 10 SSR markers at K = 2.** The vertical bar represents yam accessions assigned into two clusters.

The low number of alleles observed could suggest the moderate genetic diversity of *D. alata* accessions reported in this study. Obidiegwu et al. [34] reported 97 number of alleles in 89 *D. alata* accessions collected from nine West African countries, while Otoo et al. [32] recorded 273 number of allele in 49 *D. alata* accessions in Ghana. The different number of alleles observed between our study and others, may be due to each using different *D.alata* genotypes and different SSR markers which assess different loci.

The mean effective number of alleles per locus obtained in our study could indicate moderate genetic diversity of *D. alata* accessions due to spartial dispersion of accessions in the study sites. Different number of effective alleles were reported in previous studies. Chen et al. [17] reported 2.26 effective number of alleles using nine SSR markers in 26 *D. alata* germplasm from Southern China while Mulualem et al. [35] reported 1.71 effective number of alleles using 10 SSR markers in 33 yam accessions collected from Ethiopia. Generally, the differences in the number of effective alleles can reflect the genetic diversity of the population. The greater the value of the number of effective alleles the higher the genetic diversity of the population [36].

The mean Shannon diversity index observed in this study indicates moderate genetic diversity among *D. alata* populations collected from different regions. The Shannon diversity value from this study is higher compared to 0.78 reported by Cao et al. [37] in *D. alata* accessions from China and lower than 1.29 observed by Siqueira et al. [38] in *D. alata* cultivars from Brazil. As a vegetatively propagated crop, yam tends to maintain a high level of heterozygosity [39, 40]. However, due to the presence of few flowers and dioecism, limited genetic recombination may occur in yam as reported by Wu-Wenqiang et al. [41], which may explain the moderate genetic diversity observed in this study.

The mean expected heterozygosity (the genetic diversity) observed in our study suggests moderate genetic diversity of *D. alata* accessions. The sporadic and narrow distribution of *D. alata* accessions from the study sites could also account for the observed results. The genetic diversity from study is low compared to the 0.66 reported by Arnau et al. [16] in *D. alata* accessions collected from South Pacific, Asia, Africa and the Caribbean. Similarly, Otoo et al. [32] revealed higher (0.77) genetic diversity in *D. alata* accessions collected from Ghana than the genetic diversity reported in this study. The limited and inconsistence flowering in yam could explain different levels of genetic diversity reported in previous studies compared to our study. In our study, few accessions flowered which may affect the level of genetic diversity observed. Several factors such as planting materials used (tuber or seed) and environmental factors, such as photoperiod have been reported to affet the rate and pattern of flowering [42]. Furthermore, the different levels of genetic diversity could results from geographical patterns of species and life-history [43]. In our study, accessions were collected from regions with sporadic geographical distribution with possibility of both outcrossing and selfing mode of reproduction.

The inbreeding coefficient ($F_{IS}$) showed that three markers (30%) had a negative inbreeding coefficient, indicating an excess of heterozygosity within the population. This suggests the possible hybridization of these yam accessions. Similar results by Mulualem et al. [35] stated an excess of heterozygotes in 3 loci out of 10 SSR markers used in yam accession collected from Ethiopia. Similarly, Mengesha et al. [44] observed that 3 of 7 SSR markers had an excess of the heterozygotes in Guinea yam accessions collected from Ethiopia. The observed excess heterozygosity implies the presence of multiple demes within the population [44]. In our study, the private alleles were higher in Kagera region and lowest in Lindi region. Private alleles indicate the existence of unique genes that have evolutionary significance. *D. alata* accession from the Kagera region showed the highest number of private alleles, higher genetic diversity and higher Shannon diversity index. Therefore, Kagera region is rich in yam diversity compared to the rest of the studied regions.

The AMOVA results in this study revealed that the highest variation was attributed to the within individual variation than among population. The low variation observed among the population in our study suggests a high rate of germplasm exchange between neighbor regions. It also indicates that there is no significant geographical differentiation within *D. alata* accessions from the study sites. High variation within individuals could be due to recombination, mutation, cross-pollination, or other processes that can produce new genes and alleles [45]. A similar study by Siqueira et al. [31] revealed higher variation within the population (95.91%), while among the population, was only 4.09%. Similarly, Loko et al. [46] reported that 96% of the variation was within population while 4% was attributed to among population. The high within population variation observed implies a lack of genetic structure.

The principal coordinate analysis showed the dispersion of *D. alata* accessions into two groups regardless of their geographical origin. Most accessions from different geographical distributions were clustered within the same groups. The clustering pattern suggests that these *D. alata* accessions are genetically similar despite different local names and geographical origins. The clustering observed in our study could be due to the lack of improved yam varieties which causes farmers to grow similar varieties over the years. Similar results were reported by Wu-Wenqiang et al. [41] while assessing the genetic diversity of 142 *D. alata* cultivars from South China. The authors reported only two groups despite collecting *D. alata* cultivars from eight provinces in China. Asfaw et al. [47] revealed that grouping of accessions with different names into same group could be due to similar accessions that might be known by different names in different regions.

The genetic distance observed in this study was highest for accessions sampled from Kilimanjaro and Mtwara. Nei [48] reported that genetic distance is mostly related to geographic distance. The highest genetic distance observed between Kilimanjaro and Mtwara can be due to less possibility of germplasm exchange between the two regions. Mtwara is in the Southern part of Tanzania, while Kilimanjaro is in the North, and the distance apart is about 1,100 km. However, the high genetic distance observed in this study between the two regions may enhance the possibilities of yam improvement through breeding [49].

The UPGMA cluster analysis grouped 63 *D. alata* accessions into two distinct clusters (A and B) and four sub-clusters. Despite different local names, the dendrogram grouped *D. alata* accessions irrespective of their geographical origin, except for sub-cluster B1, which contained *D. alata* accessions from Kagera region. The grouping observed in our study suggests these accessions are genetically similar. In contrast, grouping in sub-cluster B1 could suggest that these accessions might have some unique genetic characteristics compared to the rest of the groups. Despite yam accessions being collected from five regions, the two groups observed in this study suggest lack of wide genetic diversity. The exchange of planting materials between farmers over the years also explains the pattern. Similar studies in Brazil, Benin and Ethiopia reported similar results, in which no geographical pattern was observed among yam accessions [31, 35, 50].

The structure analysis confirmed the two clusters generated by UPGMA and showed no clear structure of *D. alata* accessions. The analysis revealed a high admixture rate of *D. alata* accessions between the five sampling regions. The high proportion of admixture observed could be due to the high frequency of exchange of yam genetic resources among farmers from neighboring regions and gene flow among *D. alata* accessions [14, 32]. A similar study by Siqueira et al. [31] reported admixture between *D. alata* accessions collected from Brazil and attributed the presence of admixture accessions to germplasm exchange between different regions.

## Conclusion

This study revealed moderate genetic diversity of *D. alata* accessions from six major growing regions of Tanzania. The study showed that *D. alata* accessions are grouped into two clusters, despite they are originated from different geographical regions, which suggests their genetic similarities. The lack of a yam breeding program in the country and extensive farmers' exchange of planting materials may have contributed to the moderate genetic diversity. Information obtained from this study is crucial for selecting *D. alata* accessions for breeding programs and conservation strategies.

## Acknowledgments

We thank Ms. Magreth Lupembe of Tanzania Agricultural Research Institute and other staff from Tanzania Agricultural Research Institute at Maruku, Selian and Naliendele for their technical support during the field survey. The authors acknowledge the International Institute of Tropical Agriculture (Tanzania), especially the Molecular lab unit, for their support during laboratory work. We also appreciate the assistance of Ms. Lucy Muthui of the International Livestock Research Institute—SegoliP unit and Dr. Inosters Nzuki of Africa Biosystems Limited, Nairobi, Kenya for the SSR data scoring.

## Author Contributions

**Conceptualization:** Joseph Innocent Massawe, Gladness Elibariki Temu.

**Data curation:** Joseph Innocent Massawe.

**Formal analysis:** Joseph Innocent Massawe, Gladness Elibariki Temu.

**Investigation:** Joseph Innocent Massawe.

**Methodology:** Joseph Innocent Massawe, Gladness Elibariki Temu.

**Supervision:** Gladness Elibariki Temu.

**Validation:** Gladness Elibariki Temu.

**Writing – original draft:** Joseph Innocent Massawe.

**Writing – review & editing:** Joseph Innocent Massawe, Gladness Elibariki Temu.

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
