## [Decision Letter · Decision Letter 0]

10 Jan 2023

PONE-D-22-32759Unravelling the genetic diversity of water yam (Dioscorea alata L.) accessions from Tanzania using simple sequence repeat (SSR) markersPLOS ONE

Dear Dr. Massawe,

Thank you for submitting your manuscript to PLOS ONE. After careful consideration, we feel that it has merit but does not fully meet PLOS ONE’s publication criteria as it currently stands. Therefore, we invite you to submit a revised version of the manuscript that addresses the points raised during the review process.

We look forward to receiving your revised manuscript.

Kind regards,

Mehdi Rahimi, Ph.D.

Academic Editor

PLOS ONE

Journal Requirements:

4. We note that Figure 1 in your submission contain map images which may be copyrighted. All PLOS content is published under the Creative Commons Attribution License (CC BY 4.0), which means that the manuscript, images, and Supporting Information files will be freely available online, and any third party is permitted to access, download, copy, distribute, and use these materials in any way, even commercially, with proper attribution. For these reasons, we cannot publish previously copyrighted maps or satellite images created using proprietary data, such as Google software (Google Maps, Street View, and Earth). For more information, see our copyright guidelines: http://journals.plos.org/plosone/s/licenses-and-copyright.

(1) You may seek permission from the original copyright holder of Figure 1 to publish the content specifically under the CC BY 4.0 license.  

5. Please upload a new copy of Figure 3 and 4 as the detail is not clear. Please follow the link for more information: https://blogs.plos.org/plos/2019/06/looking-good-tips-for-creating-your-plos-figures-graphics/

https://blogs.plos.org/plos/2019/06/looking-good-tips-for-creating-your-plos-figures-graphics/

Reviewers' comments:

Reviewer's Responses to Questions

**Comments to the Author**

1. Is the manuscript technically sound, and do the data support the conclusions?

Reviewer #1: Yes

Reviewer #2: Yes

Reviewer #3: No

2. Has the statistical analysis been performed appropriately and rigorously? 

Reviewer #1: Yes

Reviewer #2: No

Reviewer #3: Yes

3. Have the authors made all data underlying the findings in their manuscript fully available?

Reviewer #1: Yes

Reviewer #2: Yes

Reviewer #3: Yes

4. Is the manuscript presented in an intelligible fashion and written in standard English?

Reviewer #1: Yes

Reviewer #2: No

Reviewer #3: Yes

5. Review Comments to the Author

Reviewer #1: The present study has been done to estimate the genetic diversity present in the Water yam (Dioscorea alata L.). 63 accessions from six different regions of Tanzania were collected and analyzed using 10 SSR markers. In this study no geographical isolation was found and the accessions shared genetic similarity with each other. The genetic diversity study of Water yam (Dioscorea alata L.) has been done with different research groups using different molecular markers such as RAPD, ISSR, AFLP. SSR and SNP markers. Since this is the first report on genetic diversity of Water yam (Dioscorea alata L.) from Tanzania therefore this study may be interesting for researchers working on this crop.

Reviewer #2: Line 36 add “cluster analysis” before UPGMA

In Table 1 accession on #54 & 55 are repetitions

Line 130 what do you mean by number of different alleles? Do you mean number of polymorphic alleles?

AMOVA only assess the genetic diversity and genetic relationship among populations not accessions

Why do you use a burn-in –period of 10 000? Some researchers use large burn-in-period.

Line 158-160: Shannon information index is a population parameter not locus parameters. It does not has any biological meaning in the case of markers i.e. Shannon information index value for Dab2E07 was 1.47, what does this implies?

Line 173: Some of the regions were not represented by sufficient samples such as Arusha (2) and Kilimanjaro (4), hence the genetic parameter values were under estimated. I suggest to either merging them together or the values should be adjusted based on number of samples per population.

Table 5: why among accession variation was not included? GenAlex can give the results even though the percentage variation is 0%. But I doubt that it won't be the case?

Figure 2: the classification of the accession is confusing. Based on PCoA 1 & 2 the accessions are classified into three potential groups. If we consider PCoA1, then the A & B classification should be completely different. So please recheck it.

Line 222: It is also ideal to report on the genetic distance between accessions since there may be a chance that the same accession given different name in different regions.

Table 7: The table legend is not informative. Specify the values as genetic distance and identity based on above and below diagonal for ease of reading

Line 135-136: How is the relationship in terms of number and patterns of clustering between the two clustering approaches i.e. PCA and the dendrogram generated using UPGMA

Figure 4: The quality of Figure 4 should be improve using Excel.

Citation style when references added at the beginning, in the middle and at the end of the sentence should be rechecked throughout the paper

Line 276: Is not clear? was the low number of alleles due to low or high allelic variants?

Line 276-278: the 97 and 273 values indicated are not the number of alleles per locus, but they are the total number of allele observed overall population.

Line 303-304: Is flowering time and pattern affected by environmental factors in Yam? If so, explain the results obtained in this study and previous studies based on environmental suitability for flowering.

Line 316-318: This might be attributed by the large number of samples collected from Kagera regions unless the values are adjusted based on number of sample per region. This is misleading.

Line 342-343: The implication is not correct since Arusha is represented by only two accessions.

Line 353-354: The results revealed the lack of wide genetic diversity but not lack of improved variety. The molecular data is all about variation at the DNA sequence level.

Reviewer #3: The MS entitled “Unravelling the genetic diversity of water yam (Dioscorea alata L.) accessions from Tanzania using simple sequence repeat (SSR) markers” have been critically reviewed and comments are as follows:

In the present investigation, the authors evaluated genetic diversity among 63 accessions of the water yam using 10 SSR markers and reported moderate level of genetic diversity. The genetic diversity in this crop has been widely examined at various locations and using large number of accessions and more advance marker systems such as SNP. The present investigation does not provide any new information’s and the number of accessions and number of markers are very less. Large number of markers evenly distributed across the genome are recommend to be used to assessment of genetic diversity. Therefore, the present study could not be recommended for publication in PLOS One.

6. PLOS authors have the option to publish the peer review history of their article (what does this mean?). If published, this will include your full peer review and any attached files.

Reviewer #1: **Yes: **Dr Rakesh Singh

Reviewer #2: **Yes: **Amelework B Assefa

Reviewer #3: **Yes: **Hemant Kumar Yadav

---

## [Author Response · Author response to Decision Letter 0]

20 Mar 2023

Editor comment: All editor comments are well addressed in the attached manuscript.

Reviewer 1: We appreciate your comments and appreciation to our manuscript.

Reviewer 2: All comments and suggestions raised by the reviewer were addressed point by point and incorporated in the revised clean manuscript.

Reviewer 3: Thank you very much for the time devoted in reviewing our manuscript and your comment. Currently, SSR markers are still markers of choice in genetic diversity studies of different crops. However, given the availability of funding, more advanced markers will be employed. On the other hand, this study provided information about the genetic diversity of D. alata accessions that are grown in Tanzania. This information is currently lacking in Tanzania, however, the information is very important in breeding and conservation strategies.

---

## [Decision Letter · Decision Letter 1]

3 Apr 2023

PONE-D-22-32759R1Unravelling the genetic diversity of water yam (Dioscorea alata L.) accessions from Tanzania using simple sequence repeat (SSR) markersPLOS ONE

Dear Dr. Massawe,

Thank you for submitting your manuscript to PLOS ONE. After careful consideration, we feel that it has merit but does not fully meet PLOS ONE’s publication criteria as it currently stands. Therefore, we invite you to submit a revised version of the manuscript that addresses the points raised during the review process.

We look forward to receiving your revised manuscript.

Kind regards,

Mehdi Rahimi, Ph.D.

Academic Editor

PLOS ONE

Journal Requirements:

Additional Editor Comments (if provided):

Dear Author

The reviewer(s) have recommended minor revisions to your manuscript. Therefore, I invite you to respond to the reviewer(s)' comments and revise your manuscript.

With Thanks

Reviewers' comments:

Reviewer's Responses to Questions

**Comments to the Author**

1. If the authors have adequately addressed your comments raised in a previous round of review and you feel that this manuscript is now acceptable for publication, you may indicate that here to bypass the “Comments to the Author” section, enter your conflict of interest statement in the “Confidential to Editor” section, and submit your "Accept" recommendation.

Reviewer #1: All comments have been addressed

Reviewer #2: All comments have been addressed

2. Is the manuscript technically sound, and do the data support the conclusions?

Reviewer #1: Yes

Reviewer #2: Yes

3. Has the statistical analysis been performed appropriately and rigorously? 

Reviewer #1: Yes

Reviewer #2: Yes

4. Have the authors made all data underlying the findings in their manuscript fully available?

Reviewer #1: Yes

Reviewer #2: No

5. Is the manuscript presented in an intelligible fashion and written in standard English?

Reviewer #1: Yes

Reviewer #2: Yes

6. Review Comments to the Author

Reviewer #1: The reply to reviewers comments point-wise has not been submitted by authors. This needs to be submitted. The corrections in the manuscript needs to be done in track change mode which is missing. Therefore, authors are requested to submit the revised manuscript in the track change mode for further evaluation.

Reviewer #2: The abstract needs to be revisited. They have changed the analysis and the results were different from the original analysis, for example, the AMOVA results have been changed as among population, among individual and within individual. I have also seen few editorial issues to be addressed.

7. PLOS authors have the option to publish the peer review history of their article (what does this mean?). If published, this will include your full peer review and any attached files.

Reviewer #1: **Yes: **Rakesh Singh

Reviewer #2: **Yes: **Amelework B. Assefa

---

## [Author Response · Author response to Decision Letter 1]

2 May 2023

Response to editor comment:

Reference list has been thoroughly checked, the list is complete and correct. No retracted paper has been cited in this manuscript.

Response to reviewer 1:

Point-wise reviewer’s comments were submitted in matrix form during previous revision. Further, the revised manuscript with track changes was also submitted in the previous revision. Both attachments are still available in the Editorial manager.

Response to reviewer 2:

The abstract has been updated accordingly. AMOVA and other parameters have been corrected.

---

## [Editor Report · Decision Letter 2]

17 May 2023

Unravelling the genetic diversity of water yam (Dioscorea alata L.) accessions from Tanzania using simple sequence repeat (SSR) markers

PONE-D-22-32759R2

Dear Dr. Massawe,

We’re pleased to inform you that your manuscript has been judged scientifically suitable for publication and will be formally accepted for publication once it meets all outstanding technical requirements.

Kind regards,

Mehdi Rahimi, Ph.D.

Academic Editor

PLOS ONE
---

## [Editor Report · Acceptance letter]

22 May 2023

PONE-D-22-32759R2 

Unravelling the genetic diversity of water yam (*Dioscorea alata* L.) accessions from Tanzania using simple sequence repeat (SSR) markers 

Dear Dr. Massawe:

I'm pleased to inform you that your manuscript has been deemed suitable for publication in PLOS ONE. Congratulations! Your manuscript is now with our production department. 

Kind regards, 

on behalf of

Associate Prof. Mehdi Rahimi 

Academic Editor

PLOS ONE